# Silicic and Humic Acid Priming Improves Micro- and Macronutrient Uptake, Salinity Stress Tolerance, Seed Quality, and Physio-Biochemical Parameters in Lentil (*Lens culinaris* spp. *culinaris*)

**DOI:** 10.3390/plants12203539

**Published:** 2023-10-11

**Authors:** Deepak Rao, Sangita Yadav, Ravish Choudhary, Dharmendra Singh, Rakesh Bhardwaj, Sharmistha Barthakur, Shiv Kumar Yadav

**Affiliations:** 1Division of Seed Science and Technology, ICAR—Indian Agricultural Research Institute, New Delhi 110012, India; dr.deepakrao13@gmail.com (D.R.); ravianu1110@gmail.com (R.C.); 2Division of Genetics, ICAR—Indian Agricultural Research Institute, New Delhi 110012, India; dharmenderpbg@rediffgmail.com; 3ICAR—National Bureau of Plant Genetic Resources, Pusa Campus, New Delhi 110012, India; rakesh.bhardwaj1@icar.gov.in; 4ICAR—National Institute of Plant Biotechnology, New Delhi 110012, India

**Keywords:** abiotic stress, antioxidant, seed quality parameters, ROS, histochemical, lentil

## Abstract

Lentil is an important grain legume crop which is mostly grown on marginal soils that hamper its productivity. Improvement of salt tolerance in lentils is considered to be a useful strategy of utilizing salt-affected lands in an economic manner. This study was conducted to evaluate the effectiveness of seed priming using silicic acid and humic acid both seperately and in combination to improve salt stress tolerance among three different lentil varieties: IPL-316 (tolerant), PSL-9, and PDL-1 (susceptible). The concentrations and durations of treatments were standardized under the normal condition and the salinity stress condition. Salt stress hindered seedling emergence and biomass production and accelerated Na^+^ toxicity and oxidative damage at the seedling stage in untreated seeds. Nevertheless, chemical priming improved early seedling emergence, increased root length, shoot length, and seed vigor index I and II, and reduced the mean germination time. A significant quantitative change in biochemical parameters under normal and salinity stress conditions was observed in IPL-316,viz. Specifically, for IPL-316, the following parameters were observed (values under the normal condition and values under salt stress conditions, respectively): chlorophyll-a (16 and 13 mg/g Fw), chlorophyll-b (25 and 16 mg/g FW), total chlorophyll content (42 and 30 mg/g FW), relative leaf water content (92% and 82%), total soluble sugars (26 and 33 ug/g FW), free amino acid (10 and 7 mg/g FW), total phenol (26 and 24 mg of GAE/g FW), total protein (35 and 29 mg/g FW), carbohydrate (208 and 173 mg/g FW), superoxide dismutase (SOD) (29 and 35 unit/min./g FW), proline (0.28 and 0.32 u mol/g FW), catalase (CAT) (84 and 196 unit/mL/g FW), and peroxidase (POX) (217 and 738 unit/mL/g FW). Furthermore, histochemical analysis of H_2_O_2_ and O^2−^, micronutrients, and macronutrients also increased, while malondialdehyde (MDA) (0.31 and 0.47 nmol/mL FW) content decreased using silicic and humic acid priming under salt stress conditions. The combination of silicic and humic acids improved seedling growth and reduced oxidative damage in lentil plants under salt stress conditions. The combination of silicic and humic acid priming hastened seedling emergence, seed quality parameters, and biochemical parameters under salt stress over respective control. To the best of our knowledge, this is the first report of integrated chemical priming in lentils for salinity stress. In conclusion, chemical priming using a combination of silicic and humic acid performed better in terms of seed quality due to enhanced antioxidant machinery, better membrane stability and osmolyte protection, and enhanced nutrient uptake under salt stress conditions.

## 1. Introduction

Lentil is one of the most nutritious grain legumes and it is primarily cultivated for its uses as human food and animal feed. It thrives well in semi-arid environments, often encountering abiotic stresses like drought, salinity, etc. Global climate change has been increasing soil salinity, which could be a major threat to lentil production globally [1]. The FAO Land and Plant Nutrition Management Service found that over 6% of the world’s land is affected by salinity. Out of the current 230 million hectares of irrigated land, 45 million hectares are salt-affected (19%), and of the 1500 million hectares under dry land agriculture, 32 million (2%) are salt-affected to varying degrees. One out of every five irrigated lands is affected by salinity, resulting in the annual loss of 1.5 million ha of agricultural lands which subsequently lose their suitability for agricultural production. If these conditions continue to move in this direction, 50% of cultivated lands will be at risk of loss by 2050 [2]. The rise in groundwater levels with high salt content, coupled with inefficient drainage and irrigation systems and the overuse of fertilizers, are responsible for soil salinity [3,4]. Salinity causes a 12–100% yield reduction in grain legumes, attributed to the suppression of physiological and morphological attributes [5,6]. It also affects reproductive growth, limiting the number of flowers and pollen functions, resulting in fewer pods with smaller grains [7]. The reduction in growth and development of plants under salt stress is mainly due to osmotic stress, nutrient imbalance, and specific ion toxicity that trigger the excessive production of reactive oxygen species (ROS) [8,9]. Enhancing salinity tolerance in plants is an effective strategy for ensuring successful and economical crop production in salt-affected soils [10,11].

Poor stand establishment is a serious challenge for successful crop legume production on salt-affected soils, where germination and seedlings are the most susceptible to salt stress [12]. Salt stress reduces and delays seedling germination with scanty seedling growth and biomass production due to Na^+^ and Cl^−^ ion toxicity or due to osmotic stress on germinating seeds [13,14]. Hence, early emergence and a better crop stand ensure successful crop production under salt stress [15]. Seed priming is an effective technique for improving germination and seedling establishment under salinity stress. Several plant growth regulators, organic compounds, inorganic salts, plant-growth-promoting rhizobacteria (PGPR), arbuscular mycorrhizal fungi (AMF), and wood vinegar (WV) antibacterial agents have been employed for priming, thereby bolstering the salinity tolerance in grain legumes [16,17,18,19,20,21,22]. Likewise, several reports have narrated the role of seed priming application in improving the performance of various crop plants under suboptimal growth conditions [23,24]. Salinity-induced oxidative stress escalates the presence of reactive oxygen species (ROS) in plant cells. The toxic impact of ROS is lipid peroxidation, causing membrane deterioration, DNA damage, and protein impairment [25]. The effect of salinity on plant growth unfolds in two phases: the first phase is dominated by osmotic stress and the second by ion-specific stress with a subsequent generation of free radicles and reactive oxygen species (ROS) [20].

Annually, 1–2% of cultivable land is reduced due to soil salinity, and worldwide, about 800 million hectares (23%) of total arable land are affected by this issue [26]. In light of this, the present study aims to assess the impact of seed priming utilizing silicic acid, humic acid, or a combination of both to enhance salt stress tolerance in lentil.

## 2. Results

The effect of priming treatments on seed quality parameters in lentils under salinity stress (100 mM) and normal conditions exhibited significant differences in all the analyzed seed quality traits.

### 2.1. Seed Quality Parameters

Under non-saline conditions (control), seed priming, involving a combination of silicic and humic acid at low concentrations, yielded significantly higher germination percentages and all other assessed seed quality parameters. Similarly, when subjected to a 100 mM salinity stress condition, the combination treatment exhibited significantly better results compared to other priming treatments as well as control for all the seed quality and physio-biochemical parameters. The findings regarding the effect of priming treatments on various seed quality parameters of lentils have been given below.

#### 2.1.1. Germination Percentage and Mean Germination Time (MGT)

In the present study, the average germination percentage was found to be significantly high in the IPL-316 variety (92% and 88%), followed by PSL-9 and PDL-1 (82%, 85% each), in the control and salinity stress conditions, respectively (Figure 1A). The application of chemical priming with silicic acid and humic acid under the salinity stress conditions significantly improved the germination percentage. This means that it increased by 4% each after hydropriming, silicic acid, and humic acid priming. In comparison, it increased by 7% after seed priming using a combination of silicic acid and humic acid under the normal condition in IPL-316. Similarly, in PSL-09, the germination percentage increased by 12%, 5%, 9%, and 11% after hydropriming, followed by silicic acid, humic acid, and a combination of humic acid and silicic acid, respectively, under the normal condition. Similarly, for PDL-01, the germination percentage was found to be increased by 1% through hydropriming, 6% with silicic acid, 12% with humic acid, and 14% using a combination of humic acid and silicic acid as compared to the control across all three varieties. Similar trends were observed under the salinity stress condition. The combination treatment showed significantly better results with all three varieties. The germination percentage in IPL-316 increased significantly by 3% after hydropriming, 5% with silicic acid, 8% with humic acid, and 11% using a combination of humic acid and silicic acid priming treatment as compared to the control. Similarly, in the PSL-9 under the salinity stress conditions, the germination percentage increased by 4%, 6%, 3%, and 4% after hydropriming, silicic acid, humic acid, and the combination of humic acid and silicic acid, respectively. For PDL-01, the germination percentage was found to be significantly better results and increased by 5% after hydropriming, 6% using silicic acid, 6% using humic acid, and 9% using a combination of humic acid and silicic acid priming treatments.

The mean germination time (MGT) was found to be significantly low in the PDL-1 variety (1.01 d), followed by IPL-316 (1.13 d), which was on par with the PSL-9 variety (1.17 d). The average mean germination time was found to be the lowest when seeds were primed with combination of silicic and humic acid (1.00 d) under both the normal and the salinity stress condition, while it was found to be the highest in the control (1.32 d) (Figure 1B). In comparison, it decreased by 25% after hydropriming, 28% after silicic acid, 21% after humic acid priming, and 27% after seed priming using a combination of humic acid and silicic acid under the normal condition in IPL-316. Similarly, in PSL-09, the MGT decreased significantly by 21%, followed by 18%, 14%, and 31% after hydropriming, silicic acid, humic acid and a combination of humic acid and silicic acid, respectively, under the normal condition. Similarly, for PDL-01, the MGT was found to be decreased significantly by 25% after hydropriming, 11% while using silicic acid, 14% with humic acid, and 31% using a combination of humic acid and silicic acid as compared to the control.

Under the salinity stress condition, the combination treatment showed significantly better results compared to the other treatments among all three varieties. This means that it was found to be decreased significantly in IPL-316 (19%) after hydropriming, 21% using silicic acid, 22% using humic acid, and 29% using a combination of humic acid and silicic acid as compared to the control. Similarly, in the PSL-9 under the salinity stress condition, the MGT decreased by 15%, 10%, 2%, and 2% after hydropriming followed by silicic acid, humic acid, and a combination of humic acid and silicic acid treatments, respectively. In the case of PDL-01, the MGT was found to yield significantly better results and decreased by 45% after hydropriming, 38% using silicic acid, 29% by humic acid, and 41% using a combination of humic acid and silicic acid priming treatments.

#### 2.1.2. Seed Vigor Indexes

Seed vigor index-II was found to be the highest after seed priming using a combination of humic acid and silicic acid (6) under both the normal and the salinity stress conditions, as compared to the control (4). Mean seed vigor index-II was found to be the highest in IPL-316 (7 and 6), followed by PSL-9 (6 and 5) and PDL-1 (4 and 4) under normal and stress conditions, respectively (Figure 1C). IPL-316 exhibited significant improvements in seed vigor index-II by 4% with hydropriming, 6% with silicic acid, and 9% with humic acid priming, while it was significantly increased by 19% after seed priming using a combination of humic acid and silicic acid under the normal condition. For PSL-09, seed vigor index-II was found to be increased significantly by 4%, 7%, 10%, and 17% as compared to the control after hydropriming followed by silicic acid, humic acid, and a combination of humic acid and silicic acid, respectively, under the normal condition. Similarly, significant enhancements were also observed for seed vigor index-II for PDL-01 after the priming treatments as compared to the control. It was found to be increased significantly in comparison with the control: 23% with hydropriming, 33% using silicic acid, 41% with humic acid, and 44% after treating the seeds using a combination of humic acid and silicic acid.

A similar trend of results was observed under the salinity stress condition, using a combination treatment proving superior among all three varieties. Under the 100 mM salinity stress condition, seed vigor index-II was found to be increased significantly by 2% after hydropriming, 6% with silicic acid, 14% with humic acid, and 27% using a combination of humic acid and silicic acid in IPL-316 as compared to the control. Similarly, under the salinity stress condition in PSL-9, seed vigor index-II was found to be increased significantly by 4%, 8%, 11%, and 20% after hydropriming followed by silicic acid, humic acid, and using a combination of humic acid and silicic acid, respectively. For PDL-01, similar observations were also found and showed better results with respect to seed vigor index-II that increased significantly by 22% with hydropriming, 30% using silicic acid, 36% with humic acid, and 38% using a combination of humic acid and silicic acid.

Seed vigor index-I was found to be the highest after priming the lentil seeds using a combination of humic acid and silicic acid (2138) under both the normal and the salinity stress conditions as compared to the control (1648). Among the varieties, IPL-316 exhibited maximum seed vigor index-I (2324 and 1742) followed by PSL-9 (2155 and 1510) and PDL-1 (2026 and 1381) under normal and stress conditions, respectively (Figure 1D). Seed vigor index-I was found to be increased significantly in IPL-316 by 7% after hydropriming, 12% using silicic acid, 13% using humic acid, and 15% after seed priming using a combination of humic acid and silicic acid under the normal condition. For PSL-09, seed vigor index-I was found to be increased significantly by 4%, 8%, 15%, and 19% after hydropriming followed by silicic acid, humic acid, and using a combination of humic acid and silicic acid, respectively, under the normal condition. For PDL-01, seed vigor index-I was found to be increased significantly by 24% using hydropriming, 32% using silicic acid, 36% using humic acid, and 42% using a combination of humic acid and silicic acid, as compared to the control.

A similar trend was observed under the salinity stress condition in which the combination treatment showed better results among all three varieties. Seed vigor index-I was found to be increased significantly by 6% after hydropriming, 12% using silicic acid, 17% using humic acid, and 19% using a combination of humic acid and silicic acid in the IPL-316 variety under the 100 mM salinity stress condition, as compared to the control. In PSL-9, seed vigor index-I was found to be increased significantly by 4%, 8%, 15%, and 20% after hydropriming followed by silicic acid, humic acid, and using a combination of humic acid and silicic acid, respectively. Similarly, in PDL-01, seed vigor index-I was found to be significantly increased by 23% after hydropriming, 31% using silicic acid, 34% using humic acid, and 40% using a combination of humic acid and silicic acid.

#### 2.1.3. Speed of Germination and Germination Rate Index

The speed of germination was found to be the highest after seed priming using a combination of humic acid and silicic acid under both the normal and the salinity stress conditions (23) as compared to the control (18). The IPL-316 variety exhibited the highest speed of germination (23 and 21) followed by PSL-9 (18 and 21) and PDL-1 (21 and 20) under the normal and stress conditions, respectively (Figure 2A). In PSL-09, the speed of germination was found to be increased significantly by 12%, 11%, 17%, and 20% after hydropriming, silicic acid, humic acid, and using a combination of humic acid and silicic acid, respectively, under the normal condition. For PDL-01, the speed of germination was found to be increased significantly by 13% using hydropriming, 14% by silicic acid, 23% using humic acid, and 25% using a combination of humic acid and silicic acid as compared to the control.

Similar trends were observed under the salinity stress conditions where the combination treatment showed better results among all three varieties. In IPL-316, the speed of germination was found to be increased significantly by 18% after hydropriming, 27% using silicic acid which was on par with the humic acid treatment and showed a 28% increase, while it increased significantly by 35% using a combination of humic acid and silicic acid under the 100 mM salinity stress condition as compared to the control. For PSL-9 under the salinity stress condition, the speed of germination was found to be increased significantly by 11%, 11%, 16%, and 11% after hydropriming, silicic acid, humic acid, and using a combination of humic acid and silicic acid, respectively. Similarly, in PDL-01, the speed of germination was increased BY up to 16% after hydropriming, 20% with silicic acid, 25% using humic acid, and 27% using a combination of humic acid and silicic acid.

The germination rate index (GRI) was found to be the highest after seed priming using a combination of humic acid and silicic acid under both the normal and the salinity stress conditions (91%) as compared to the control (71%). The IPL-316 variety exhibited a high average germination rate index (GRI) (88%) followed by PDL-1 (81%) and PSL-9 (78%) under the normal and the stress conditions, respectively (Figure 2B). The germination rate index was found to be increased significantly with the priming treatments, viz. 22%, after hydropriming, 24% using silicic acid, and 24% using humic acid priming, while it increased significantly by 26% after seed priming using a combination of humic acid and silicic acid under the normal condition. In PSL-09, the germination rate index was found to be increased significantly by 9%, 9%, 14%, and 10% after hydropriming, silicic acid, humic acid, and using a combination of humic acid and silicic acid, respectively, under the normal condition. For PDL-01, the germination rate index increased significantly by 12% after hydropriming, 16% using silicic acid, 22% using humic acid, and 25% using a combination of humic acid and silicic acid as compared to the control.

Under the salinity stress condition, the combination treatment showed better results among the three varieties. In IPL-316, the germination rate index was found to be increased significantly by 3% after hydropriming, 11% with silicic acid, 12% by humic acid, and 18% using a combination of humic acid and silicic acid under 100 mM salinity stress condition as compared to the control. For PSL-9 under the salinity stress condition, the germination rate index increased significantly by 11%, 11%, 16%, and 20% after hydropriming, silicic acid, humic acid, and using a combination of humic acid and silicic acid, respectively. Similarly, in the PDL-01, it was also increased significantly by 16% in hydropriming, 20% using silicic acid, 25% using humic acid, and 27% using a combination of humic acid and silicic acid.

#### 2.1.4. Germination Index

The germination index (GI) was found to be the highest after seed priming using a combination of humic acid and silicic acid under both the normal and the salinity stress conditions (232) as compared to the control (207). The IPL-316 variety exhibited a high average germination index (232) followed by PSL-9 (214) and PDL-1 (213) under normal and stress conditions (Figure 2C). The germination index was found to be increased significantly in IPL-316 by 2% after hydropriming, 4% using silicic acid, and 5% using humic acid priming, while it increased significantly by 6% after seed priming using a combination of humic acid and silicic acid under the normal condition. In PSL-09, the germination index was found to be increased significantly by 3%, 6%, 9%, and 11% after hydropriming followed by silicic acid, humic acid, and a combination of humic acid and silicic acid, respectively, under the normal condition. Similarly, for PDL-01, the germination index was increased by 1% after hydropriming, 6% using silicic acid, 10% using humic acid, and 12% using a combination of humic acid and silicic acid, as compared to the control.

Under the 100 mM salinity stress condition, the germination index in IPL-316 was found to be increased significantly by 2% after hydropriming, 7% with silicic acid, 7% using humic acid, and 10% using a combination of humic acid and silicic acid as compared to the control. In PSL-9, the germination index was found to be increased significantly by 5%, 6%, 12%, and 9% after hydropriming followed by silicic acid, humic acid, and a combination of humic acid and silicic acid, respectively, under the salinity stress condition. For PDL-01, a similar trend was found, and the results showed germination index was increased by 1% after hydropriming, 6% by silicic acid, 11% using humic acid, and 12% using a combination of humic acid and silicic acid.

### 2.2. Physio-Biochemical Parameter under Salinity Stress and Normal Condition (Non-Saline Condition)

Our tests revealed significant differences for the biochemical traits exhibited by lentil seedlings under salt stress conditions. The experiment was performed under non-saline (normal condition) and saline conditions (100 mM). Seed priming with different combinations and concentrations of humic acid and silicic acid also showed significant differences in all the analyzed biochemical traits.

#### 2.2.1. Malondialdehyde (MDA)

The MDA content in the IPL-316 variety was found to be the highest after seed priming using a combination of humic acid and silicic acid (0.31 and 0.47 nmol/mL/fresh weight) as compared to the control (0.63 and 0.79 nmol/mL/fresh weight) under both the normal and salinity stress conditions, respectively (Figure 3A). In the IPL-316 variety, MDA content was decreased significantly by 30% each after hydropriming and silicic acid, 44% using humic acid, and 51% using a combination of humic acid and silicic acid under the normal condition, while it was decreased significantly by 24% each with hydropriming and silicic acid, 34% after humic acid, and 41% using a combination of humic acid and silicic acid under the salinity stress condition. For PSL-9, a similar trend was found and results showed that MDA content was decreased significantly by 47% with hydropriming, 52% with silicic acid, 57% using humic acid, and 69% using a combination of humic acid and silicic acid under the normal condition, while it was decreased significantly by 30% after hydropriming, and by 33% using a combination of silicic acid and humic acid under the salinity stress condition. The significantly higher decrement (39%) was observed when seeds were primed using a combination of humic acid and silicic acid under the salinity stress condition. Similarly, the PDL-1 variety also yielded significant results after priming: the MDA content was found to be decreased significantly by 21% after hydropriming, 26% with silicic acid, 26% with humic acid, and by 38% using a combination of humic acid and silicic acid under normal and salinity stress conditions.

#### 2.2.2. Superoxide Dismutase (SOD)

The SOD content was found to be the highest after seed priming using a combination of humic acid and silicic acid (29 and 35 unit/min/g fresh weight) as compared to the control (30 and 18 unit/min/g fresh weight) under both the normal and salinity stress conditions, respectively (Figure 3B). Overall, the superoxide dismutase content increased significantly when the seeds were primed using a combination of humic acid and silicic acid. In the IPL-316 variety, it increased significantly by 0.35% after hydropriming, 16% with silicic acid, 22% using humic acid, and 19% using a combination of humic acid and silicic acid under the normal condition; meanwhile, the SOD content increased significantly by 22% with hydropriming, 125% with silicic acid, 137% after humic acid, and 183% using a combination of humic acid and silicic acid under the salinity stress condition. For PSL-9, a similar trend was observed: SOD content was increased by 2% after hydropriming, 18% with silicic acid, 23% using humic acid, and 17% using a combination of humic acid and silicic acid under the normal condition, while the SOD content increased significantly by 15% after hydropriming, 10% with silicic acid, 27% using humic acid, and 22% using a combination of humic acid and silicic acid under the salinity stress condition. Similarly, the PDL-1 variety also showed significant differences after seed priming in which the SOD content was found to be increased significantly by 23% after hydropriming, 4% with silicic acid, 9% using humic acid, and 3% using a combination of humic acid and silicic acid under the normal condition. In the case of the salinity stress condition, the SOD content increased significantly by 4% after hydropriming, 15% with silicic acid, 15% using humic acid, and 17% using a combination of humic acid and silicic acid under the salinity stress condition.

#### 2.2.3. Catalase (CAT) and Peroxidase (POX) Content

The catalase (CAT) content was found to be the highest after seed priming using a combination of humic acid and silicic acid (84 and 196 units/mL/g/fresh weight, respectively) as compared to the control (37 and 142 units/mL/g/fresh weight) in the IPL-316 variety under both the normal and salinity stress conditions (Figure 3C). In the IPL-316 variety, catalase (CAT) content increased significantly by 35% after hydropriming, 103% with silicic acid, 92% with humic acid, and 127% using a combination of humic acid and silicic acid under the normal condition, while it was increased by 5% after hydropriming, 33% with silicic acid, 29% after humic acid, and 37% using a combination of humic acid and silicic acid under the salinity stress condition. In the PSL-9 variety, a similar pattern was observed in which catalase content was increased significantly by 10% after hydropriming, 26% with silicic acid, 41% with humic acid, and 62% using a combination of humic acid and silicic acid under the normal condition; meanwhile, CAT content increased significantly by 7% after hydropriming, 17% with silicic acid, 25% using humic acid, and 33% using a combination of humic acid and silicic acid under the salinity stress condition. Similarly, in the PDL-1 variety, it increased significantly by 23% after hydropriming, 27% with silicic acid, 30% using humic acid, and 36.91% using a combination of humic acid and silicic acid under the normal condition. Meanwhile, in the case of the salinity stress condition, the catalase content increased significantly by 14% after hydropriming, 15.73% with silicic acid treatment, 17% with humic acid, and 20% using a combination of humic acid and silicic acid.

The peroxidase (POX) content was found to be the highest after seed priming using a combination of humic acid and silicic acid (217 and 738 units/mL/g/fresh weight) as compared to the control (149 and 544 and 142 units/mL/g/fresh weight) under both the normal and salinity stress conditions, respectively (Figure 3D). The peroxidase (POX) content in IPL-316 increased significantly by 28% after hydropriming, 29% with silicic acid, 12% using humic acid, and 469% using a combination of humic acid and silicic acid under the normal condition; meanwhile, it increased significantly by 21% after hydropriming, 35% with silicic acid, 33% using humic acid, and 36% using a combination of humic acid and silicic acid under the salinity stress condition. In the PSL-9 variety, a similar kind of pattern was observed where peroxidase content increased significantly by 58% after hydropriming, which was on par with silicic acid (57%), 706% using humic acid, and 68% using a combination of humic acid and silicic acid under the normal condition; meanwhile, POX content increased significantly by 14% after hydropriming, 21% with silicic acid, 28% with humic acid, and 33% using a combination of humic acid and silicic acid under the salinity stress condition. The PDL-1 variety also showed significant results after seed priming and revealed that POX content increased significantly by 3% after hydropriming, 3% with silicic acid, 7% by humic acid, and 11% after seed priming using a combination of humic acid and silicic acid under the normal condition. In the case of the salinity stress condition, the peroxidase content increased significantly by 5% after hydropriming, 12% with silicic acid treatment, 12% with humic acid, and 19% using a combination of humic acid and silicic acid under the salinity stress condition.

#### 2.2.4. Total Phenol

The total phenolic content was found to be the highest after seed priming using a combination of humic acid and silicic acid (26 and 24 mg of GAE/g fresh weight, respectively) as compared to the control (19 and 16 mg of GAE/g fresh weight, respectively) in the IPl-316 variety under both the normal and the salinity stress conditions, respectively (Figure 4A).

The total phenolic content increased significantly in IPL-316 by 6.94% after hydropriming, 7% with silicic acid, 11% with humic acid, and 42% after a combination of humic acid and silicic acid under the normal condition, while it increased significantly by 47% after hydropriming, 22% with silicic acid, 33% using humic acid, and 46% using a combination of humic acid and silicic acid under the salinity stress condition. In the PSL-9 variety, the total phenolic content increased significantly by 26%, 34%, 35%, and 35% after hydropriming, silicic acid, humic acid, and using a combination of humic acid and silicic acid, respectively, under the normal condition; meanwhile, it was found to be increased significantly by 28% after hydropriming, 61% with silicic acid, 51% by humic acid, and 52% using a combination of humic acid and silicic acid as compared to the control under the salinity stress condition. Similarly, in PDL-1, significant variation was observed after seed priming under the normal and stress conditions. Under the normal condition, it was observed that total phenolic content increased significantly by 34% after hydropriming, 48% with silicic acid, 39% using humic acid, and 55% using a combination of humic acid and silicic acid. In the case of the salinity stress condition, the total phenol content was increased by 5% after hydropriming, 35% with silicic acid, 34% with humic acid, and 35% using a combination of humic acid and silicic acid under the salinity stress condition.

#### 2.2.5. Total Soluble Sugar (TSS)

The total soluble sugar (TSS) was found to be the highest after seed priming using a combination of humic acid and silicic acid (26 and 33µg/g fresh weight, respectively) as compared to the control (10 and 17 µg/g fresh weight, respectively) in the IPL-316 variety under both the normal and the salinity stress conditions, respectively (Figure 4B).

The TSS content increased significantly with the priming treatments as compared to the control. In IPL-316, TSS increased by 86% after hydropriming, 124% with silicic acid, 147% using humic acid, and 154% after a combination of humic acid and silicic acid under the normal condition; meanwhile, it increased by 39% with hydropriming, 75% with silicic acid, 88% after humic acid, and 92% using a combination of humic acid and silicic acid under the salinity stress condition. In the PSL-9 variety, a similar pattern was observed for TSS content: it increased significantly by 42% after hydropriming, 47% with silicic acid priming, 53% using humic acid, and 56% after seed priming using a combination of humic acid and silicic acid under the normal condition. Meanwhile, TSS content increased significantly by 46% after hydropriming, 9% with silicic acid, 12% after humic acid, and 60% after seed priming using a combination of humic acid and silicic acid under the salinity stress condition. In the PDL-1 variety, the results were found to be significant after seed priming as compared to the control. TSS content increased significantly by 18% after hydropriming, 22% with silicic acid, 30% using humic acid, and 51% after seed priming using a combination of humic acid and silicic acid under the normal condition. However, in the case of the salinity stress condition, the TSS content increased significantly by 36% after hydropriming, 43% with silicic acid, 49% using humic acid, and 54% after seed priming using a combination of humic acid and silicic acid under the salinity stress condition.

#### 2.2.6. Carbohydrate Content

Carbohydrate content was found to be the highest after seed priming using a combination of humic acid and silicic acid (208 and 173 mg/g/fresh weight, respectively) as compared to the control (84 and 484 mg/g/fresh weight, respectively) in the IPL-316 variety under both the normal and salinity stress conditions, respectively (Figure 4C).

Carbohydrate content increased significantly with seed priming as compared to the control. In IPL-316, it was increased after hydropriming (26% and 45%) followed by silicic acid (60% and 105%), humic acid (118% and 205%), and using a combination of humic acid and silicic acid (147% and 255%) under the normal and the 100 mM salinity stress conditions, respectively. For PSL-9, the carbohydrate content increased significantly after hydropriming (11% and 18%) followed by silicic acid (42% and 55%), humic acid (43.32% and 55.61%), and using a combination of humic acid and silicic acid (49.76% and 61.88%) under the normal and salinity stress conditions, respectively. Similarly, in PDL-1, the carbohydrate content was also increased significantly after hydropriming (4% and 4%), followed by silica acid (27% and 27%), humic acid (41% and 40%), and using a combination of humic acid and silicic acid (68% and 53%) under normal and salinity stress conditions, respectively.

#### 2.2.7. Total Protein and Free Amino Acid Content

Protein content was found to be the highest after seed priming using a combination of humic acid and silicic acid (35 and 29 mg/g/fresh weight, respectively) as compared to the control (19 and 12 mg/g/fresh weight, respectively) in the IPL-316 variety under both the normal and salinity stress conditions, respectively. The average protein content was found to be the highest in the IPL-316 variety (32 mg/g/fresh weight) followed by PLS-9 (26 mg/g/fresh weight) and PDL-1 (26 mg/g/fresh weight) (Figure 4D).

The total protein content increased significantly with the seed priming treatment as compared to the control. In IPL-316, it was increased after hydropriming (9% and 32%) followed by silicic acid (13% and 44%), humic acid (15% and 75%), and using a combination of humic acid and silicic acid (16% and 142%) under normal and salinity stress conditions, respectively. In PSL-9, the protein content also increased significantly after hydropriming (42% and 17%), followed by silicic acid (47%), humic acid (53%), and using a combination of humic acid and silicic acid (56%) under normal and salinity stress conditions, respectively. Similarly, for PDL-1, it increased significantly after hydropriming (32% and 16%), silica acid (18% and 67%), humic acid (30% and 63%), and using a combination of humic acid and silicic acid (49% and 65%) under normal and 100 mM salinity stress conditions, respectively.

The free amino acid content was found to be the highest after seed priming using a combination of humic acid and silicic acid (10 and 7 mg/g fresh weight, respectively) as compared to the control (4 and 2 mg/g fresh weight, respectively) in the IPL-316 variety under both the normal and salinity stress conditions, respectively (Figure 4E).

Free amino acid content increased significantly after seed priming treatment as compared to the control. In IPL-316, it increased after hydropriming (31% and 27%), silicic acid (98% and 237%), humic acid (135% and 95%), and using a combination of humic acid and silicic acid (155% and 222%) under normal and salinity stress conditions, respectively. In PSL-9, the free amino acid content increased significantly with hydropriming (10% and 53%), silicic acid (12% and 66%), humic acid (16% and 68%), and using a combination of humic acid and silicic acid (35% and 74%) under the normal and salinity stress conditions, respectively. Similarly, in the PDL-1 variety, it was also found to be increased significantly after hydropriming (36% and 29%), silicic acid (42% and 33%), humic acid (39% and 67%), and using a combination of humic acid and silicic acid (75 % and 69%) under normal and salinity stress conditions, respectively.

#### 2.2.8. Proline Content

The proline content was found to be the highest after seed priming using a combination of humic acid and silicic acid (0.31, 0.32, and 0.32 units/mL/g/fresh weight, respectively) as compared to the control (0.25 and 0.24 units/mL/g/fresh weight, respectively) in the IPL-316 variety under both the normal and salinity stress conditions, respectively (Figure 4F).

The proline content increased significantly with the seed priming treatment as compared to the control. In IPL-316, it increased significantly after hydropriming (4% and 12%), silicic acid (16% and 37%), humic acid (20% and 44%), and using a combination of humic acid and silicic acid (24% and 55%) under the normal and stress conditions, respectively. In PSL-9, the proline content increased significantly with hydropriming (6% and 0%), silicic acid (15% and 16%), humic acid (27% and 26%), and using a combination of humic acid and silicic acid (40% and 45%) under the normal and salinity stress conditions, respectively. Similarly, in the PDL-1 variety, it was also increased significantly after hydropriming (21% and 43%), silica acid (26% and 53%), humic acid (43% and 62%), and using a combination of humic acid and silicic acid (55% and 68%) under the normal and stress conditions, respectively.

#### 2.2.9. Chlorophyll-a, -b, and Total Chlorophyll Content

Chlorophyll-a was found to be the highest after seed priming using a combination of humic acid and silicic acid (16 and 13 mg/g FW) as compared to the control (12 and 10 mhg/g FW) in the IPL 316 variety under the normal and salinity stress conditions, respectively (Figure 5A).

Chlorophyll-a content increased significantly with the seed priming treatment as compared to the control. In IPL-316, it was increased after hydropriming (29% and 5%), silicic acid (30% and 11%), humic acid (32% and 21%), and using a combination of humic acid and silicic acid (33% and 28%) under normal and stress conditions, respectively. In PSL-9, chlorophyll-a content was also increased significantly with hydropriming (12% and 1%), silicic acid (41% and 9%), humic acid (38% and 12%), and using a combination of humic acid and silicic acid (42% and 13%) under the normal and salinity stress conditions, respectively. Similarly, in the PDL-1 variety, it increased significantly after hydropriming (4% and 0%), silicic acid (20% and 12%), humic acid (37% and 17%), and using a combination of humic acid and silicic acid (78% and 19%) under normal and stress conditions, respectively.

Chlorophyll-b was found to be the highest after seed priming using a combination of humic acid and silicic acid (25 and 16 mg/g FW) as compared to the control (12 and 10 mg/g FW) in the IPL 316 variety under the normal and the salinity stress conditions, respectively (Figure 5B).

Chlorophyll-b content increased significantly with the seed priming treatment as compared to the control. In IPL-316, it increased significantly after hydropriming (58% and 31%), silicic acid (68% and 37%), humic acid (95% and 42%), and combination of humic acid and silicic acid (102% and 60%) under normal and salinity stress conditions, respectively. In PSL-9, the chlorophyll-b content increased significantly with hydropriming (11% and 4%), silicic acid (46% and 4%), humic acid (59% and 10%), and using a combination of humic acid and silicic acid (59% and 15%) under the normal and salinity stress conditions, respectively. Similarly, in the PDL-1 variety, it increased significantly after hydropriming (27% and 0.43%), silica acid (32% and 1%), humic acid (33% and 3%), and a combination of humic acid and silicic acid (73% and 15%) under the normal and salinity stress conditions, respectively.

The total chlorophyll content was found to be the highest after seed priming using a combination of humic acid and silicic acid (42 and 30 mg/g FW) as compared to the control (25 and 20 mg/g FW) in the IPL 316 variety under the normal and the salinity stress conditions, respectively (Figure 5C).

The total chlorophyll content increased significantly with the seed priming treatment as compared to the control. In IPL-316, it increased significantly after hydropriming (44% and 18%), silicic acid (50% and 24%), humic acid (64% and 32%), and a combination of humic acid and silicic acid (69% and 43%) under the normal and salinity stress conditions, respectively. In PSL-9, the total chlorophyll content increased significantly with hydropriming (12% and 3%), silicic acid (64% and 11%), humic acid (64% and 11%), and using a combination of humic acid and silicic acid (69% and 14%) under the normal and salinity stress conditions, respectively. Similarly, in the PDL-1 variety, it increased significantly after hydropriming (17% and 0.26%), silicic acid (26% and 6%), humic acid (35% and 11%), and using a combination of humic acid and silicic acid (72% and 17%) under the normal and stress conditions, respectively.

#### 2.2.10. Relative Water Content

The relative water content (RWC) was found to be the highest after seed priming using a combination of humic acid and silicic acid (92 and 82 mg/g FW) as compared to the control (73 and 64 mg/g FW) in the IPL 316 variety under the normal and the salinity stress conditions, respectively (Figure 5D).

The relative water content increased significantly with seed priming treatment as compared to the control. In IPL-316, it was increased after hydropriming (21% and 20%), silicic acid (23% and 31%), humic acid (27% and 32%), and using a combination of humic acid and silicic acid (26% and 27%) under the normal and stress conditions, respectively. In PSL-9, the relative water content increased significantly after hydropriming (8% and 12%), silicic acid (8% and 17%), humic acid (7% and 18%), and using a combination of humic acid and silicic acid (6% and 13%) under the normal and salinity stress conditions, respectively. Similarly, in the PDL-1 variety, it increased significantly after hydropriming (12% and 9%), silicic acid (28% and 9%), humic acid (25% and 30%), and using a combination of humic acid and silicic acid (27% and 30%) under normal and stress conditions, respectively.

### 2.3. Histochemical Detection of O^2−^ and H_2_O_2_ by NBT and DAB in Lentil Seedlings

After eighteen hours of radicle emergence, seeds with a 2 mm radicle length were treated with DAB and NBT, showing as brown and blue, respectively. It was observed that a brown color indicated H_2_O_2_ accumulation after the staining with a 3, 3′-diaminobenzidine (DAB) solution, while purple-blue staining was indicative of O^2−^ production using nitro blue tetrazolium (NBT) solution in both the normal and salinity stress conditions. The color intensity was observed in the control and primed seeds. In this study, the maximum level of super oxide radicles was observed when seeds were primed using a combination of humic acid and silicic acid (T5), while it was found to be the lowest in control seeds (T1) under both the normal and salinity stress conditions (Figure 6). Similarly, the hydrogen peroxide accumulation activity was found to be the highest in the seeds after priming using a combination of humic acid and silicic acid (T5), while it was found to be the lowest in control seeds (T1) under both the normal and salinity stress conditions (Figure 7). The detection of H_2_O_2_ and O^2−^ with DAB and NBT, respectively, was further validated in the spectro-microscopic analysis, which ensured the effect of seed priming using a combination of humic acid and silicic acid (T5). It was observed that the radicles (after 18 h) were darkly stained with brown (DAB) and purple-blue (NBT) color using a combination of humic acid and silicic acid (T5), while control seeds displayed very light staining in radicles under both the normal and salinity stress conditions, respectively (Figure 6 and Figure 7).

### 2.4. Macronutrient (Ca, Mg) and Micronutrient (Fe, Zn) Analysis

In the present study, micro- and macronutrients were analyzed in all three varieties of lentil after different chemical priming approaches under normal and salinity stress conditions (Table 1 and Table 2). Significant differences were observed after the priming treatments in all three varieties.

#### 2.4.1. Macronutrient (Ca and Mg) Content

Calcium (Ca) content was found to be the highest when seeds were primed using a combination of humic acid and silicic acid (T5) (1198.93 mg/kg and 983 mg/kg), while it was found to be the lowest in the control (T1) (982 mg/kg and 784 mg/kg) under normal and stress conditions, respectively (Table 1 and Table 2). Similarly, magnesium (Mg) content also increased significantly after seed priming as compared to the control under both the normal and salinity stress conditions. The magnesium (Mg) content was found to be the highest in T5 (1224 mg/kg) and the lowest was observed in the control (1124 mg/kg) under the normal condition (Table 1). A similar trend was observed under the 100 mM salinity stress condition, in which the Mg content was found to be the highest when seeds were primed using a combination of humic acid and silicic acid (T5) (1214 mg/kg); meanwhile, it was found to be the lowest in the control (T1) (998 mg/kg) (Table 2).

#### 2.4.2. Micronutrient (Fe and Zn) Content

Iron (Fe) content was found to be the highest when seeds were primed using a combination of humic acid and silicic acid (T5) in the normal (20 mg/mL) (Table 1) and stress conditions (19 mg/mL) (Table 2), while it was found to be the lowest in the control (T1) (14 mg/mL and 12 mg/mL), under normal and stress conditions, respectively. Similarly, zinc (Zn) content also increased significantly after seed priming as compared to the control under both the normal and salinity stress conditions. The zinc content was found to be the highest in T5 (1.44 mg/mL) and the lowest was observed in the control (T1) (1.11 mg/mL) under the normal condition (Table 1). Under the 100 mM salinity stress condition, a similar trend was observed and the results showed that the zinc content was found to be the highest when seeds were primed using a combination of humic acid and silicic acid (T5) (1.21 mg/mL), while it was found to be the lowest in the control (0.78 mg/mL) (Table 2).

## 3. Discussion

Mismanagement of agriculture, leading to increased salt concentrations in soils, has severe consequences for agricultural productivity, as it reduces crop growth and survival [27]. Recent studies have provided evidence of significant decreases in crop yields under saline conditions, specifically in wheat and tomato crops [28,29]. Consequently, addressing the issues of soil salinity is crucial if we are to meet the food demands of a growing global population [30]. Research to determine the most efficient ways of reducing food scarcity is necessary. One such solution is to increase the tolerance of lentil crop to salinity stress, increasing the overall yield.

The application of humic acid and silicic acid is one potential remedy now being looked into since it can enhance crop development by influencing inherent plant features. By preventing cell division and hence reducing cell elongation and growth, excessive salt concentrations in soils can have a deleterious effect on plant shape. This results from nutritional abnormalities, oxidative damage, and osmotic stress. In soil, silicon dioxide (SiO_2_) is the most prevalent form of silicon (Si). All forms of Si have low solubility and limited mobility in the soil. The primary soluble forms of Si are poly- and monosilicic acids; monosilicic acid is weakly adsorbed and exhibits low mobility within soil.

Increasing the concentration of monosilicic acid in the soil solution allows plants to absorb micro- and macronutrients from the soil profile. This increase occurs because phosphate and silicate anions share a chemical resemblance, leading to a competitive reaction in the soil. However, the insolubility of monosilicic acid is slightly reduced through interactions with heavy metals, iron, aluminum, and manganese [31].

Studies focusing on the response of lentil to salt stress have indicated that seed priming with silicic acid and humic acid can improve the seed quality parameters and seedling vigor [32]. In the current investigation, the effects of seed priming with a silicic and humic acid on germination percentage (GP), shoot length (SL), root length (RL), seed vigor index-I (SVI-I), and seed vigor index-II (SVI-II), MGT, SOG, GRI, and GI, under the normal and salinity stress conditions (100 mM), were explored in different lentil varieties. A seed priming strategy was discovered to improve these parameters for all lentil varieties under salinity stress, as well as to aid in the prevention of antioxidant damage caused by salinity stress. Seed priming is widely known for repairing membrane damage caused by seed storage or abiotic stresses [33]. Previous research has found that seed priming causes biochemical changes in seeds, such as the activation of enzymes involved in cellular metabolism, inhibition of metabolism, dormancy breaking, and water imbibition, which aids the germination process [34,35].

Seed priming is a pre-sowing technique that involves soaking seeds in different substances to improve germination and early seedling growth. Silicic acid, humic acid, and their combinations have gained popularity as seed priming agents due to their positive effects on plant growth and development. Silicic acid, also known as soluble silicic acid, is a mineral element that plays a vital role in plant growth and development. It has been shown to boost seed germination, plant growth, and tolerance to biotic and abiotic stressors. For example, seed priming with silicic acid enhanced seedling development and germination rates in wheat plants exposed to salty environments [36]. Similarly, it was also reported that silicic acid priming enhanced early growth and salt stress resistance in pea, wheat, and rice seedlings [37].

Humic acid, a complex organic compound derived from the decomposition of plant and animal materials, is widely utilized as a soil conditioner and fertilizer. Its advantages include improving soil structure, increasing nutrient availability, and stimulating plant development. The administration of humic acid by seed priming improved seed germination rate and seedling growth in drought-stressed maize plants [38]. Similarly, it has been reported that humic acid priming also enhances both the seed germination rate and early growth in onion seedlings [39]. The combination of humic acid and silicic acid as seed priming agents resulted in enhancement of seed germination rate, increased seedling growth, and improved the nutrient uptake for wheat plants growing under saline conditions [40]. Furthermore, it was discovered that a combination of humic acid and silicic acid improved the seed germination rate and promoted early growth in cucumber seedlings [41].

The present study revealed significant variations in seedling characteristics based on different priming practices. Among the various applications and control plants, hydropriming and priming with 3 mM silicic acid and 600 ppm humic acid demonstrated superior effectiveness. Chemical priming was found to expedite the germination process by facilitating rapid water absorption by the seeds. Priming practices initiate germination through the activation of the biochemical machinery in the seed, contributing to enzyme production, cell wall expansion, and breaking of dormancy. Silicic acid, an essential bioactive element, plays a crucial role in enhancing leaf morphology, root penetration, stress tolerance, plant growth, resistance to pathogens, and nutrient uptake [42,43]. It also forms silicic acid deposits in the roots, reducing apoplastic flow and absorption of toxic minerals, thereby reducing the water loss through transpiration [44]. The promotion of lateral root formation by priming applications is a significant outcome for seedling development. The lateral roots are critical components of the plant’s overall root system design, which includes all underground organs, and play an important role in water and nutrient intake [45,46].

Under abiotic stress conditions, the activity of enzymes that degrade chlorophyll rises, resulting in the degradation of chlorophyll pigments [47]. Salinity stress specifically causes the degradation of chlorophyll pigments, impairing the photosynthesis process [48]. Salinity stress has a negative impact on the chloroplast, the most sensitive organelle, resulting in damage to photosynthetic pigments in diverse crops [49]. The degeneration of chlorophyll is believed to be associated with the production of reactive oxygen species (ROS), and it reduces the photosynthesis rate and enhances cellular respiration. In the present study, exposure to salinity-induced stress negatively impacted photosynthetic pigments, including chlorophyll-a, -b, and total chlorophyll content. Conversely, seed priming with silicic acid, humic acid, and a combination of silicic acid and humic acid significantly increased the levels of photosynthetic pigments in cells. Chemical priming with varying levels of silicic acid and humic acid protected the chloroplast membranes from photooxidation and created a favorable environment for efficient functioning of the photosynthetic machinery under oxidative stress conditions.

Plant status has a considerable impact on carbohydrate metabolism, with salt stress increasing soluble sugar concentration. Plants generally respond to salt by storing soluble carbohydrates [50]. The diminished activity of enzymes involved in the pentose phosphate pathway and glycolysis, such as glucose-6-phosphate dehydrogenase, glucose phosphate isomerase, and 6-phosphogluconate dehydrogenase, may be responsible for this buildup [51]. Interestingly, at higher salinity levels, there is a reduction in plant carbohydrate content. It is possible that GA_3_, a plant hormone, plays a crucial role in regulating the osmotic balance between carbohydrate metabolism and salt tolerance. In the current study, an overall increase in the total protein content was observed under 100 mMNaCl, while it was decreased under the 140 mMNaCl condition. This decrease may be attributed to the inhibitory effects of higher levels of Na^+^ and Cl^−^ on protein synthesis [52]. Salinity stress leads to the replacement of K^+^ with Na^+^, resulting in a physiological imbalance, since K^+^ is an essential component for protein synthesis and enzyme activation [53].

Total phenol and free amino acid were affected, while total free amino acids increased significantly in all the cultivars studied in this experiment of lentil under saline conditions. Similarly, it was also reported that free amino acid content was increased in wheat under saline conditions [54]. Crop plants produce free amino acids under salt stress conditions, which help maintain osmotic balance by reducing cellular osmotic potential in plants under saline circumstances [55]. Chemical priming with silicic and humic acid may have considerably impacted total free amino acids or total phenolics in both wheat cultivars under the control and the saline conditions. Contrary to expectations, seed treatment with triacontanol dramatically increased the accumulation of free amino acids and phenols in green gram under normal growth circumstances [56].

Under salt stress, the activities of SOD, MDA, and proline in squash leaves increased, while seed priming with silicic acid caused them to decrease [57]. The activities of CAT and POD, both in roots and leaves, increased under salt stress, which is beneficial for plants as they act as scavengers for free radicles [58]. Furthermore, the activities of these enzymes were further improved by the addition of adequate amounts of humic acid and silicic acids. These investigations repeatedly showed that the addition of silicic acid could lower the levels of malondialdehyde (MDA), a byproduct of lipid peroxidation [59]. This demonstrates the complexities of variations in antioxidant enzyme activity. Despite this intricacy, the moderating impact of silicic acid and humic acid on oxidative damage in salt-stressed plants has been repeatedly documented. It is crucial to note, however, that the majority of these studies were undertaken at the seedling stage and the later stages, with little information available on earlier stages. Chemical priming with silicic acid and humic acid was able to partially suppress the increase in H_2_O_2_ and O_2_ levels caused by salt stress in seedlings.

The concentrations of H_2_O_2_ and O^2−^ in *E. crusgalli* leaves were significantly higher under 2,4-D alone treatment as compared to the mild/severe salt stress treatments (4 and 8 dS/m) [60]. Combined treatments of salt and 2,4-D reduced the H_2_O_2_ and O^2−^ content significantly, except under the salt treatment comprising 8 dS m^−1^ + spray of recommended dose of 2,4-D; here, the combined treatment of 2,4-D and salt did not reduce H_2_O_2_ production in *E. crusgalli* leaves. The in situ detection of H_2_O_2_ and O^2−^ with DAB and NBT further validated the spectroscopic analysis, showing that *E. crusgalli* leaves were stained with dark brown (H_2_O_2_) and deep purple-blue (NBT) colors under a treatment of 2,4-D alone (spray of recommended dose of 2,4-D) and treatments comprising combinations of 2,4-D + severe saline stress (8 dS m^−1^ + spray of recommended dose of 2,4-D). Roots treated with salt stress alone were strongly discolored, whereas those treated with combination stress treatments were stained lightly [60].

The combination of silicic and humic acid seed priming resulted in a significant increase in mineral nutrition concentration (Fe, Zn, Ca, and Mg) in lentil seedlings under normal and salinity stress conditions in this study. The findings were consistent with previous research [61], which found that seed priming increased leaf nutrient accumulation and considerably increased seedling growth in mungbean. Therefore, seed priming using a combination of silicic and humic acid can be seen as a useful method for reducing the negative effects of salinity stress by controlling nutrient uptake, chlorophyll accumulation, enhanced antioxidant machinery with a decrease in ROS, better membrane stability with a decrease in lipid peroxidation, and increased osmolyte protection, which results in a noticeable increase in lentil seedling growth.

## 4. Material and Methods

### 4.1. Experimental Details

During the years 2022–2023, the present research was carried out in the Division of Seed Science and Technology, ICAR—Indian Agricultural Research Institute, New Delhi (India). For this experiment, three lentil crop types (IPL-316, PSL-9, and PDL-1) were obtained from the Division of Genetics at ICAR-IARI in New Delhi (India). In this study, we standardized multiple combinations and doses of humic acid and silicic acid for seed priming at various durations. We conducted five treatments: T1 = control; T2 = hydropriming; T3 = chemical priming with 3 mM silicic acid (@18 h); T4 = 600 ppm humic acid (18 h); T5 = combination of 1 mM silicic and 100 ppm humic acid (16 h). These treatments were ultimately used to assess the tolerance of seedlings against a 100 mM salinity stress level. A completely randomized design with 50 seeds/4 replications was used to set up the experimental treatments.

### 4.2. Seed Quality Parameter

The germination percentage, shoot length (cm), root length (cm), seed vigor index-I, and seed vigor index-II (ISTA 2022) were recorded on the first day (d) and the final day (d) counts of the lentil crop.

### 4.3. Mean Germination Time (MGT)

Assessment of vigor based on the germination test can be more precisely performed through a daily count of the physiological germination characteristics (i.e., radicle emergence). The MGT of seedlings was measured through the daily reading of radicle emergence until all the radicles had emerged [61].
MET = ∑ Dn/∑ n
where ‘D’ represents the number of days from the start of emergence and ‘n’ represents the number of seedlings emerged on ‘D’ days.

### 4.4. Chlorophyll Content

The samples were collected from the young seedlings and weighed. Then, they were macerated with 3.0 mL of 80% acetone after transfer into an Eppendorf tube. Next, the sample was centrifuged at 12,000 rpm for 7 min. Finally, the ELISA plate reader was used for observations at wavelengths of 470, 645, 652, and 663 nm [62].
Chla = 12.9 (Ab663) − 2.69 (Ab645) × V/1000 × W
Chlb = 22.9 (Ab645) − 4.68 (Ab663) × V/1000 × W
Total Chl = Chla + Chlb

### 4.5. Relative Water Content

The relative water content (RWC) was estimated according to the modified method [63]. Samples (0.5 g) were saturated in 100 mL distilled water for 24 h at 4 °C in the dark and their turgid weights were recorded. Then, the samples were oven-dried at 65 °C for 48 h and the dry weight of the samples was recorded. The RWC of a plant tissue is expressed as unit’s percentage and calculated by using the following equation:RWC = (FM − DM)/(TM − DM) × 100
where FM, DM, and TM were the fresh, dry, and turgid masses, respectively.

### 4.6. Total Phenols

A leaf sample weighing 0.5 g was collected and macerated with 5.0 mL of 80% ethanol followed by centrifugation at 10,000 rpm for 20 min. The reaction mixture was prepared by adding 0.5 mL of extract; 2.5 mL of 10% Folin–Ciocâlteu Reagent (FCR), 2.5 mL of 7.5% NaHCO_3_. Then, this mixture was placed in a water bath at 45 °C for 45 min. After incubation, the absorbance was observed at 765 nm. Finally, the standard graph was prepared using Gallic acid [64].

### 4.7. Total Soluble Sugar

A leaf sample weighing 0.5 gm was taken and macerated with 5.0 mL of ethanol (80%). The sample was centrifuged at 2000 rpm for 20 min; then, the reaction mixture was prepared by adding 1.0 mL of extract, 1.0 mL of phenol (5%), 5 mL of H_2_SO_4_ (96%). Next, the sample was placed in a water bath at 26–30 °C for 20 min. The yellow-orange color is an indication of the reaction having concluded, and the absorbance was recorded at 490 nm. Finally, the standard curve was prepared using glucose [65].

### 4.8. Total Carbohydrates

Initially, 100mg of seedlings was weighed and placed in boiling water in a tube, followed by the addition of 5 mL of 2.5N HCl. This was hydrolyzed in a water bath for 3 h. Then, the tubes were cooled at room temperature and neutralized with solid sodium carbonate until the effervescence ceased. The final volume was reduced to 100 mL and this was centrifuged. Then, 1 mL of supernatant and 4 mL of anthrone reagent (0.1 gm in 50 mL H_2_SO_4_) were mixed. The samples were heated for 8 min in a boiling water bath, followed by rapid cooling. Finally, the absorbance was recorded at 630 nm. Carbohydrate content in the seedling sample was calculated using the standard curve with glucose [66].

### 4.9. Proline Content

The proline content was obtained according to the modified method [67]. Initially, the fresh weight of sample was taken and macerated with 3% sulphosalicylic acid followed by centrifugation at 15,000× *g* rpm for 20 min. The reaction mixture was prepared by adding 0.5 mL of acidic ninhydrin, 0.5 mL glacial acetic acid, and 0.5 mL of extract. Then, the sample was incubated in a water bath at 95 °C for 1 h. It was then cooled at room temperature. A measure of 3 mL of toluene was added after a few minutes. Finally, readings were taken at 520 nm and a standard curve with proline was prepared.

### 4.10. Free Amino Acid

A measure of 0.5 g of seedlings were taken for free amino acid analysis according to the modified method [68]. They were macerated in a ratio of 60:25:15 *v*/*v* of methanol: chloroform: water. The samples were incubated in a water bath at 60 °C for 2 h followed by centrifugation at 5000× *g* rpm for 10 min. The reaction mixture was prepared by adding 1.0 mL of extract, 1.0 mL of 0.1 acetate buffer (pH 4.3), and 1.0 mL of 5% ninhydrin in ethanol and incubating the samples in a water bath at 95 °C for 15 min. The samples were cooled down at room temperature for few minutes and finally the OD value was observed at 570 nm. The standard curve was prepared using glycine.

### 4.11. Total Soluble Protein

Total soluble protein content was assessed according to modified method [69]. The seedlings were macerated in 5.0 mL of 10 mM phosphate buffer (PPB) (pH 7) containing 4% polyvinylpyrrolidone (PVP). This mixture was centrifuged at 12,000 rpm for 15 min. The reaction mixture was prepared with adding the 20 µL of extract, and 980 µL of Bradford reagent followed by incubation for 15 min at room temperature. Finally, the OD value was observed at 595 nm. The standard curve was prepared using bovine serum albumin.

### 4.12. Super Oxide Dismutase (SOD)

The super oxide activity was measured using the modified method [70]. Seedlings were macerated with an added 2 mL of 50 mM phosphate buffer (pH 7). The mixture was centrifuged at 13,000 rpm for 30 min. A reaction mixture was prepared by adding 50 µL of extract, 2.15 mL of 50 mM of phosphate buffer (pH 7.8), 0.2 mL of methionine, 0.2 mL of NBT, 0.2 mL of EDTA, and 0.2 mL of riboflavin solution. After shaking the test tubes, reaction mixture was placed 30 cm below florescent light for 10 min. Then, the lights were switched off and the test tubes were covered with a black cloth. Finally, the absorbance of the reaction mixture was observed at 560 nm.

### 4.13. Malondialdehyde Content (MDA)

The MDA content was measured according to the modified method [71]. The seedlings were macerated with 1 mL of 0.1% TCA followed by centrifugation at 15,000 rpm for 15 min. The reaction mixture was prepared with adding the 0.5 mL of supernatant, 2.0 mL of 0.65% TBA followed by incubating the sample in water bath at 95 °C for 30 min. The sample was centrifuged at 10,000× *g* rpm for 5 min and observation was taken at 440, 532, and 600 nm wavelengths.
MDA content = (Ab532 − Ab600) × 106/155,000

### 4.14. Antioxidant Enzyme Activities

The activity of catalase and peroxidase is measured using the modified protocols [72,73].

#### 4.14.1. Catalase

The seedlings were macerated using 2.0 mL of 0.1M phosphate buffer, pH 7.8, using a chilled mortar and pestle. This was followed by centrifugation at 15,000 rpm for 20 min. The reaction mixture was prepared by adding 0.2 mL of extract, 1mL of 0.05 M phosphate buffer (pH 7), 1 mL of H_2_O_2_, and 0.8 mL of distilled water. Finally, absorbance was taken at 240 nm for 3 min at 30 s intervals.

#### 4.14.2. Peroxidase

Seedlings were macerated using 2.0 mL of 0.1M phosphate buffer (pH 7.8) using a chilled mortar and pestle. This was followed by centrifugation at 15,000 rpm for 20 min. The reaction mixture was prepared by adding 0.2 mL of extract, 1mL of 0.1 M phosphate buffer (pH 6.1), 0.5 mL of H_2_O_2_, 0.5 mL of guaiacol, and 0.8 mL of distill water. Finally, absorbance was recorded at 470 nm for 3 min at 15 s intervals.
POX = (finial − initial) × 3 × 1000/26.6 × 3 × 1 × 0.2 × fresh weight(1)

### 4.15. Histochemical Analysis of H_2_O_2_ and O^2^ by DBT and NBT in Leaves

Productions of superoxide anion (O^2−^) and H_2_O_2_ were detected through the NBT and 3,3′-diaminobenzidine (DAB) staining methods, respectively [74].

### 4.16. Macronutrient (Ca, Mg) and Micronutrient (Fe, Zn) Analysis

Macro- (Ca and Mg) and micronutrients (Fe and Zn) were analyzed using the atomic absorption spectrum (AAS) (Agilent 240FS AA). A wet digestion procedure was used by adding strong oxidizing acids into the sample and heating so that the organic materials in the sample decomposed. The biological samples were digested using mixed acids such as a combination of nitric acid (HNO_3_), perchloric acid (HClO_4_), sulfuric acid (H_2_SO_4_), hydrochloric acid (HCl), and hydrogen peroxide (H_2_O_2_). The final volume was made to be 25 mL using distilled water. The nutrient absorption was recorded by AAS.

### 4.17. Statistical Analysis

For each treatment, 100 seedlings were used in three replications (n = 3). Standard errors (SEs) of the arithmetic means were calculated for each treatment. The data were analyzed using analysis of variance (ANOVA) and pairwise comparison among treatment. The means were determined using Tukey’s test at *p* = 0.05 using SPSS 16.0, Chicago, IL, USA.

## 5. Conclusions

In conclusion, the use of silicic acid priming and humic acid priming has been found to improve seed quality parameters and physio-biochemical parameters under salt stress. These priming approaches have been shown in studies to improve seed germination rate, seedling growth, and nutrient uptake while minimizing the deleterious effects of salinity-induced stress on plants. Furthermore, silicic acid priming and humic priming have been shown to increase antioxidant enzyme activity, lowering oxidative damage and lipid peroxidation. The combination of humic acid and silicic acid has shown synergistic effects, further enhancing the overall performance of seeds and seedlings under salinity stress. However, more research in this area is still required. Research should be conducted at a molecular level to determine the role of the biochemical relationship under different salinity stress levels. Additionally, long-term field trials could provide insights into the sustainability and practical application of silicic acid and humic acid priming in agriculture. Overall, the use of silicic and humic acid priming shows promise as a sustainable approach to improve seed quality and enhance plant performance under salinity stress. Further research on practical applications in different crops and environments will contribute to a better understanding of the potential benefits of this approach, optimizing the use of these priming techniques in agricultural practices.

## Figures and Tables

**Figure 1 plants-12-03539-f001:**
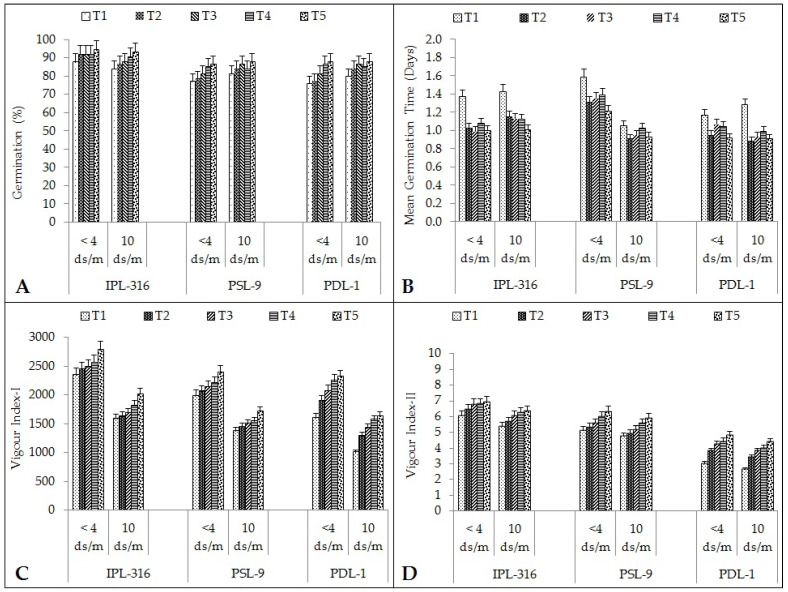
Effect of seed priming on different seed quality parameters: (**A**) germination (%); (**B**) mean germination time (days); (**C**) vigor index-I and (**D**) vigor index-II. Error bars depict significant differences (*p* < 0.05), where cases of means are three (n = 3). T1 = control; T2 = hydropriming (18 h); T3 = silicic acid (@3 mM 18 h); T4 = humic acid (@600 ppm 18 h); T5 = combination of humic acid and silicic acid (@100 ppm + 1 mM 16 h).

**Figure 2 plants-12-03539-f002:**
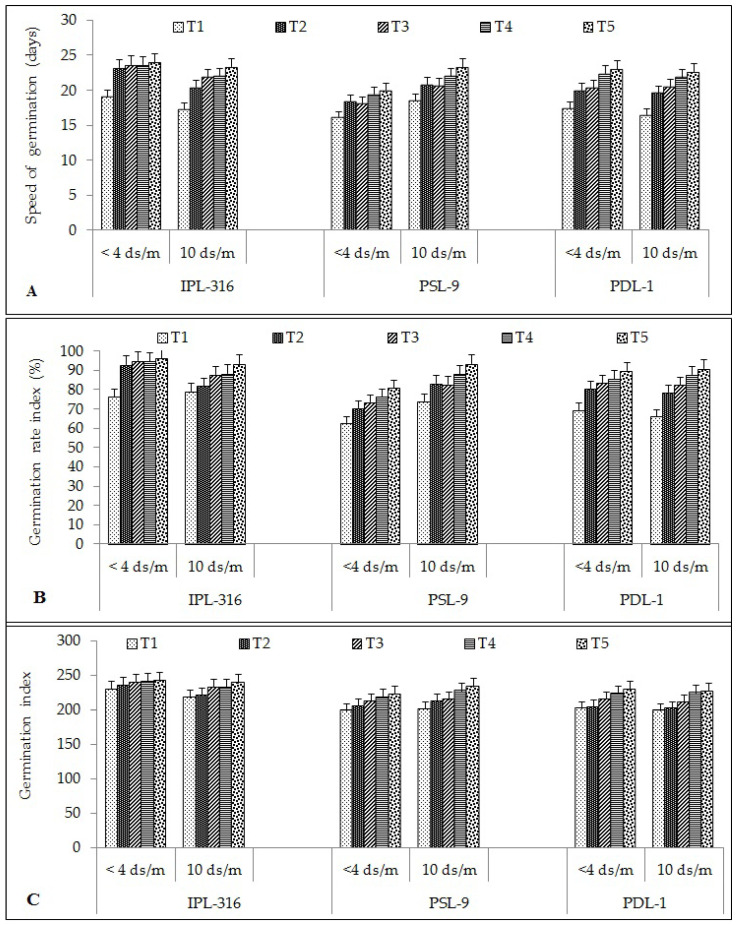
Effect of seed priming on different seed quality parameters: (**A**) speed of germination; (**B**) germination rate index; (**C**) germination index. Error bars depict significant differences (*p* < 0.05), where cases of means are three (n = 3). T1 = control; T2 = hydropriming (18 h); T3 = silicic acid (@3 mM 18 h); T4 = humic acid (@600 ppm 18 h); T5 = combination of humic acid and silicic acid (@100 ppm + 1 mM 16 h).

**Figure 3 plants-12-03539-f003:**
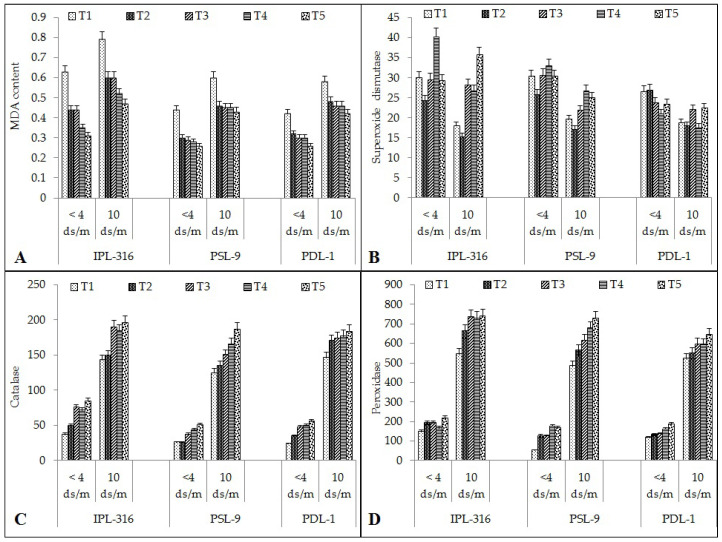
Effect of seed priming on different seed quality parameters: (**A**) malondialdehyde (MDA); (**B**) superoxide dismutase (SOD); (**C**) catalase (CAT); (**D**) peroxidase (POX). Error bars depict significant differences (*p* < 0.05), where cases of means are three (n = 3). T1 = control; T2 = hydropriming (18 h); T3 = silicic acid (@3 mM18 h); T4 = humic acid (@600 ppm 18 h); T5 = combination of humic acid and silicic acid (@100 ppm + 1 mM 16 h).

**Figure 4 plants-12-03539-f004:**
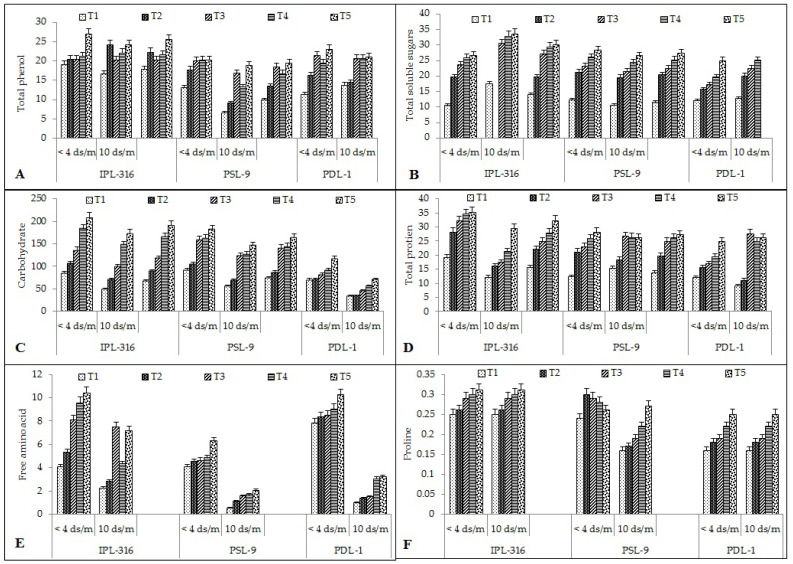
Effect of seed priming on different seed quality parameters: (**A**) total phenol; (**B**) total soluble content; (**C**) carbohydrate; (**D**) total protein; (**E**) free amino acid; (**F**) proline. Error bars depict significant differences (*p* < 0.05), where cases of means are three (n = 3). T1 = control; T2 = hydropriming (18 h); T3 = silicic acid (@3 mM 18 h); T4 = humic acid (@600 ppm 18 h); T5 = combination of humic acid and silicic acid (@100 ppm + 1 mM 16 h).

**Figure 5 plants-12-03539-f005:**
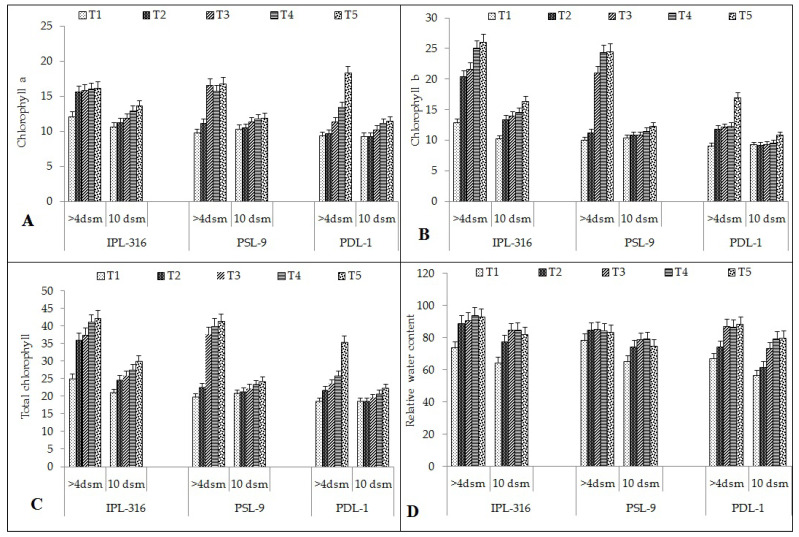
Effect of seed priming on different seed physiochemical parameters: (**A**) chlorophyll-a; (**B**) chlorophyll-b; (**C**) total chlorophyll content; (**D**) relative water content. Error bars depict significant differences (*p*< 0.05), where cases of means are three (n = 3). T1 = control; T2 = hydropriming (18 h); T3 = silicic acid (@3 mM 18 h); T4 = humic acid (@600 ppm 18 h); T5 = combination of humic acid and silicic acid (@100 ppm + 1 mM 16 h).

**Figure 6 plants-12-03539-f006:**
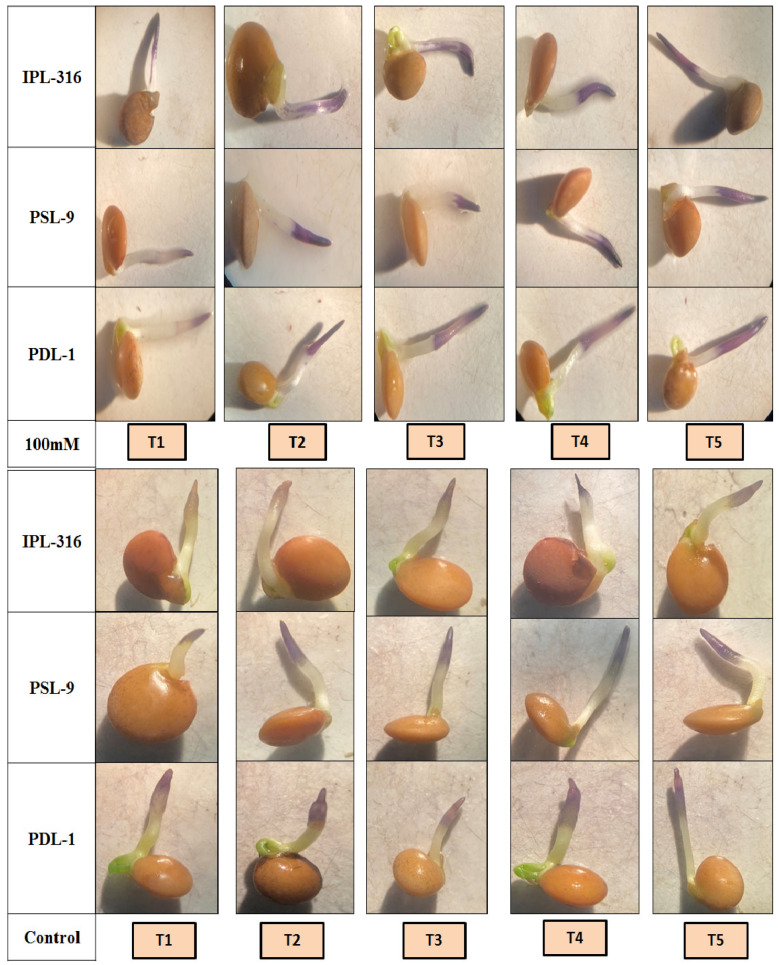
Effect of NBT for detection of super oxide radicles (O^2−^) under the control and the 100 mM salinity stress conditions. T1 = control; T2 = hydropriming (@18 h); T3 = silicic acid (@3 mM for 18 h); T4 = humic acid (@600 ppm for 18 h); T5 = combination of humic acid and silicic acid (@100 ppm + 1 mM for 16 h).

**Figure 7 plants-12-03539-f007:**
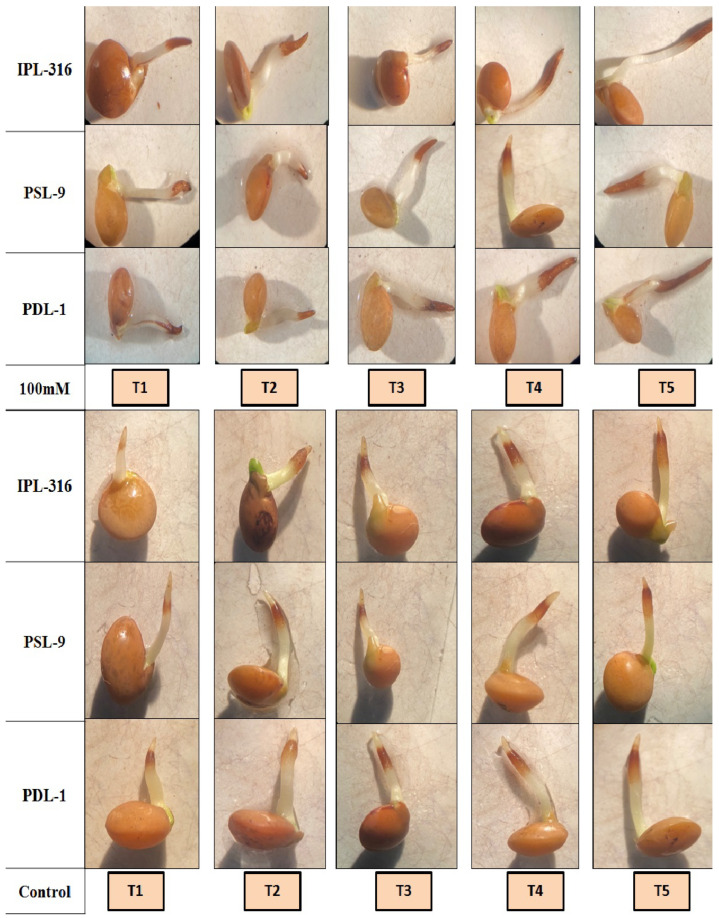
Effect of DAB for detection of H_2_O_2_ accumulation under control as well as 100 mM salinity stress conditions. T1 = control; T2 = hydropriming (@18 h); T3 = silicic acid (@3 mM for 18 h); T4 = humic acid (@600 ppm for 18 h); T5 = combination of humic acid and silicic acid (@100 ppm + 1 mM for 16 h).

**Table 1 plants-12-03539-t001:** Effects of different chemical priming on micro- and macronutrients in lentil under the normal condition.

Normal	Fe (mg/mL)				Zn (mg/mL)				Mg (mg/kg)				Ca (mg/kg)			
	V1	V2	V3	Mean	V1	V2	V3	Mean	V1	V2	V3	Mean	V1	V2	V3	Mean
T1	14.71	13.75	13.53	14.00 ^c^	1.19	0.92	1.21	1.11 ^c^	1126.10	980.50	1265.42	1124.01 ^c^	1036.45	986.35	924.45	982.42 ^c^
T2	18.23	16.48	15.76	16.82 ^b^	1.37	1.26	1.24	1.29 ^b^	1278.36	1151.48	1255.25	1228.36 ^ab^	1109.95	1031.00	994.65	1045.20 ^bc^
T3	18.34	19.37	17.79	18.50 ^ab^	1.46	1.37	1.14	1.32 ^ab^	1317.39	1182.33	1202.00	1233.91 ^ab^	1129.90	1131.90	1005.85	1089.22 ^abc^
T4	18.30	19.56	18.00	18.62 ^ab^	1.49	1.30	1.30	1.36 ^a^	1230.13	1236.14	1260.14	1242.14 ^ab^	1133.75	1137.85	1147.20	1139.60 ^ab^
T5	21.43	20.61	19.19	20.41 ^a^	1.51	1.42	1.38	1.44 ^a^	1302.39	1304.88	1272.29	1293.19 ^ab^	1186.75	1178.50	1231.55	1198.93 ^a^
Mean	18.2 ^a^	17.95 ^a^	16.85 ^a^	17.67	1.4 ^a^	1.26 ^a^	1.26 ^a^	1.31	1250.87 ^a^	1171.06 ^a^	1251.02 ^a^	1224.32 ^a^	1119.36 ^a^	1093.12 ^a^	1060.74 ^a^	1091.07
	CD(V)			1.88	CD(V)			0.33	CD(V)			112.99	CD(V)			123.93
	CD(T)			1.57	CD(T)			0.27	CD(T)			94.12	CD(T)			103.24
	CD(V × T)			2.26	CD(V × T)			0.39	CD(V×T)			135.40	CD(V × T)			135.40

Different letters superscripted on mean values depict significant differences (*p* = 0.05) among the treatments and the varieties, where mean values are three (n = 3). V1 = IPL-316; V2 = PSL-09; V3 = PDL-01; T1 = control; T2 = hydropriming (@18 h); T3 = silicic acid (@3 mM for 18 h); T4 = humic acid (@600 ppm for 18 h); T5 = combination of humic (100 ppm) and silicic acid (1 mM) for 16 h.

**Table 2 plants-12-03539-t002:** Effect of different chemical priming approaches on micro- and macronutrients in lentil under 100 mM salinity stress condition.

100 mM	Fe (mg/mL)				Zn (mg/mL)				Mg (mg/kg)				Ca (mg/kg)			
	V1	V2	V3	Mean	V1	V2	V3	Mean	V1	V2	V3	Mean	V1	V2	V3	Mean
T1	13.48	12.52	12.31	12.77 ^c^	0.87	0.60	0.89	0.78 ^c^	1007.20	851.60	1136.52	998.44 ^c^	841.65	786.55	724.65	784.28 ^c^
T2	17.00	15.25	14.53	15.59 ^ab^	1.05	0.94	1.06	0.79 ^ab^	1149.46	1022.58	1276.35	1149.46 ^ab^	915.15	831.20	794.85	847.07 ^ab^
T3	17.11	18.14	16.56	17.27 ^ab^	1.27	1.11	1.09	1.02 ^ab^	1188.49	1203.43	1273.10	1221.67 ^a^	935.10	932.10	806.05	891.08 ^ab^
T4	17.07	18.33	16.77	17.39 ^ab^	1.31	1.11	1.20	1.16 ^ab^	1151.23	1107.24	1181.24	1146.57 ^ab^	938.95	938.05	947.40	941.47 ^a^
T5	20.20	19.39	17.96	19.18 ^a^	1.32	1.36	1.29	1.21 ^a^	1173.49	1275.98	1193.39	1214.29 ^a^	989.45	978.70	981.75	983.30 ^a^
	CD(V)			1.89	CD(V)			0.36	CD(V)			125.29	CD(V)			109.82
	CD(T)			1.57	CD(T)			0.30	CD(T)			104.37	CD(T)			99.82
	CD(V × T)			2.26	CD(V × T)			0.43	CD(V × T)			150.14	CD(V × T)			131.61

Different letters superscripted on mean values depict significant differences (*p* = 0.05) among the treatments and the varieties, where cases of means are three (n = 3). V1 = IPL-316; V2 = PSL-9; V3 = PDL-1; T1= control; T2 = hydropriming (@18 h); T3 = silicic acid (@3 mM for 18 h); T4 = humic acid (@600 ppm for 18 h); T5 = combination of humic acid and silicic acid (@100 ppm + 1 mM for 16 h).

## Data Availability

Not applicable.

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
