# Peer review of "Silicic and Humic Acid Priming Improves Micro- and Macronutrient Uptake, Salinity Stress Tolerance, Seed Quality, and Physio-Biochemical Parameters in Lentil (Lens culinaris spp. culinaris)"

_plants, 2023, doi:10.3390/plants12203539_

Round 1

Reviewer 1 Report

The study is meaningful for the improvement of salt tolerance in lentil. The manuscript is well organized, and the structure and expression are fine. However, there were some issues that should be addressed in this manuscript.

1. In the abstract and conclusion sections, only qualltative results are described, some quantitative results should be supplied.

2. In Figures 1(C), (D), Figures 2(B), (C), “%” should be labeled in the vertical coordinates.

3. It is suggested that Line 120, in the title “2.1.1. Germination (%)”, “ (%)” should be deleted; line 307, in the title “2.1.7. Germination index (%) ”, “ (%)” should be deleted; the content enclosed in parentheses of section titles 4.4-4.13 (e.g. 4.4 Chlorophyll content (mg/g fresh weight), etc) should be deleted.   

4. There are too much content in the manuscript, some unnecessary content should be merged or deleted.

The language of the whole manuscript needs to be improved, e.g. lines 41-42: “Lentil is one of the most important grain legumes, mostly grown for its uses as human food and animal feed due to its high nutritious value.”, line 44: “Increased soil salinity as an outcome of global climate change”, lines 92-93, “According to the FAO Land and Plant Nutrition Management Service, over 6 % of the world’s land is affected by salinity.”, etc.

Author Response

Kindly see the file attached. Thanks

Reviewer 2 Report

In the present study “Silicic and Humic acid Priming Improves Micro and Macro Nutrients Uptake, Salinity Stress Tolerance, Seed Quality and Physio-Biochemical parameters in Lentil (Lens culinaris spp. culinaris)”, authors concluded that the chemical priming with the combination of silicic and humic acid performed better in terms of seed quality, biochemical and physiological parameters under salt stress conditions. I think that the work falls into the scope of the journal and findings are interesting, however MS demands major revision.

Comments:

Abstract: Abstract can be more concise. I would suggest to add materials and methods in Abstract. Add/change some keywords.

Introduction: There are two major concerns with this MS. First one is grammatical mistakes, language error, typographical mistakes. Second, the authors did not conceive the strong idea from review literature. Paragraphs and sentences did not have any link. There are some related articles that can help the authors e.g., Frontiers in Plant Science, 2022, 13: 973782; Microbiological Research, 2022, 266: 127254; and Industrial Crops & Products, 2022, 189: 115763.

Materials and methods: How many replications per treatment? How many plants per replication? Day/light hrs?  Humidity? Temperature? Please elaborate detail method of “Soil organic matter content”.

Results and Discussion: In results, there is a striking lack of connectors between sentences and leading to confusing. I would suggest to present your results by increase/decrease %age. Percentage should be upto two digits e.g., 13% instead of 13.4%. One way of improving Discussion is to avoid repetition of results in this part. Discussion is very shallow and need in depth discussion with the recent literature published. In discussion, there is a lack of mechanistic approach. Spellings and English language needs to be checked thoroughly. Overall, drafting of many sentences need to be improved. Tidying up the text is also suggested.

Kind regards!

In the present study “Silicic and Humic acid Priming Improves Micro and Macro Nutrients Uptake, Salinity Stress Tolerance, Seed Quality and Physio-Biochemical parameters in Lentil (Lens culinaris spp. culinaris)”, authors concluded that the chemical priming with the combination of silicic and humic acid performed better in terms of seed quality, biochemical and physiological parameters under salt stress conditions. I think that the work falls into the scope of the journal and findings are interesting, however MS demands major revision.

Comments:

Abstract: Abstract can be more concise. I would suggest to add materials and methods in Abstract. Add/change some keywords.

Introduction: There are two major concerns with this MS. First one is grammatical mistakes, language error, typographical mistakes. Second, the authors did not conceive the strong idea from review literature. Paragraphs and sentences did not have any link. Materials and methods: How many replications per treatment? How many plants per replication? Day/light hrs?  Humidity? Temperature? Please elaborate detail method of “Soil organic matter content”.

Results and Discussion: In results, there is a striking lack of connectors between sentences and leading to confusing. I would suggest to present your results by increase/decrease %age. Percentage should be upto two digits e.g., 13% instead of 13.4%. One way of improving Discussion is to avoid repetition of results in this part. Discussion is very shallow and need in depth discussion with the recent literature published. In discussion, there is a lack of mechanistic approach. Spellings and English language needs to be checked thoroughly. Overall, drafting of many sentences need to be improved. Tidying up the text is also suggested.

Kind regards!

Author Response

Kindly see the file attached. Thanks

Round 2

Reviewer 1 Report

The manuscript has been improved according to the comments and should be checked carefully.

Minor editing of English language required.